# Barriers and Facilitators of Cardiac Rehabilitation in a Middle-Income Country: A Qualitative Study from China

**DOI:** 10.3390/ijerph22040574

**Published:** 2025-04-06

**Authors:** Ying Zou, Sarah Janus, Jiamin Du, Ning Qu, Karel Zuidema, Huibert Burger, Kees Ahaus, Zhigang Guo, Sytse Zuidema

**Affiliations:** 1Department of Primary and Long-Term Care, University Medical Center Groningen, University of Groningen, 9713GZ Groningen, The Netherlands; s.i.m.janus@umcg.nl (S.J.); jiamin.du@psych.ox.ac.uk (J.D.); n.qu@umcg.nl (N.Q.); h.burger@umcg.nl (H.B.); s.u.zuidema@umcg.nl (S.Z.); 2Department of Psychiatry, University of Oxford, Oxford OX3 7JX, UK; 3Academy of Medical Engineering and Translational Medicine, Medical School, Tianjin University, Tianjin 300072, China; 4Center for Accounting, Auditing & Control, Nyenrode Business University, 3621BG Breukelen, The Netherlands; kt.zuidema@gmail.com; 5Erasmus School of Health Policy and Management, Erasmus University Rotterdam, 3062PA Rotterdam, The Netherlands; ahaus@eshpm.eur.nl; 6Department of Cardiovascular Surgery, Tianjin Chest Hospital, Tianjin 300300, China; zhigangguo@yahoo.com

**Keywords:** healthcare delivery, access to care, cardiac rehabilitation, barriers, facilitators

## Abstract

Background: Although effective and recommended by guidelines worldwide, Cardiac Rehabilitation (CR) remains scarce and underutilized. CR implementation has taken place in middle-income countries, but the progress is influenced by both positive and negative factors that remain underexplored. This study identified the barriers and facilitators of CR in a middle-income country, specifically China. Methods: An exploratory qualitative study was conducted using semi-structured interviews. Results: Fifteen CR stakeholders were interviewed. According to the interviewees, the delivery of CR is impeded due to a lack of resources, a lack of CR professionals, and a lack of coordination between health institutions. The participation of CR is hindered by a lack of awareness, a lack of reimbursement, and a lack of access to CR. However, the interviewees also mentioned facilitating factors, namely, a positive attitude of stakeholders, high motivation of some patients, and policy support. Conclusions: More awareness regarding the effectiveness of CR is needed. Implementing CR in secondary and primary health institutions could overcome the barriers regarding travel distance and transportation to faraway hospitals. The CR reimbursement methods are needed to ease the financial burden on patients. Our findings reveal factors that need to be considered by policymakers to deliver CR on a wider scale in China.

## 1. Introduction

Cardiovascular diseases (CVDs) are the main cause of mortality worldwide. Taking China as an example, CVDs caused 5.1 million deaths in 2019, accounting for more than 40% of all deaths [1,2,3]. Apart from a high risk of death, those who survive CVDs often face problems such as physical function decline, inability to work, and disability [4]. Therefore, these diseases impose a heavy socioeconomic burden on the health system, patients, and their families [4].

To ease this burden, Cardiac Rehabilitation (CR) has been applied with proven benefits in both high-income countries and low- and middle-income countries (LMICs) [5,6]. This planned program combines supervised physical activity and health education to help cardiac patients live healthier by preventing the worsening of heart disease and life-threatening events (e.g., heart attacks) [7]. In addition, CR can improve the quality of life of patients, reduce mortality, decrease the chance of rehospitalization, and increase the likelihood of returning to work [8,9]. CR consists of three phases: Phase I (inpatient phase) begins within 48 h post-operation, focusing on early mobility and risk factor education; Phase II (outpatient phase) typically lasts to 36 weeks, incorporating supervised exercise sessions; Phase III (maintenance phase) ensures long-term health stabilization, aiming to maintain the established lifestyle changes [4].

Although proven effective and recommended by global guidelines, CR remains unavailable and underutilized, with less than 40% of low- and middle-income countries (LMICs) offering CR services [10]. In these countries, the availability of CR programs is limited [11]. For example, there are only approximately two CR programs for every 100 million inhabitants in China [12].

Most studies on the reasons behind CR underutilization have been conducted in high-income countries [13,14], with only a few from middle-income countries such as Iran [11], Brazil [15], and Colombia [16]. Although the middle-income countries share a common goal of improving and enhancing the CR services delivery with high-income countries, they differ in healthcare structure and funding. Specifically, China’s healthcare system is a mixed public–private system; the national health insurance (also called basic medical insurance) serves as the foundation and covers over 95% population, while commercial health insurance provides supplementary support [17]. The healthcare institutions in China are classified into three levels: the tertiary hospitals offer high-level medical services; the secondary hospitals offer comprehensive healthcare services; and the primary hospitals or health institutions offer primary healthcare services [18].

To achieve a comprehensive understanding of the complex CR context, it is important to incorporate the perspectives from multiple stakeholders [16,19]. Rangel-Cubillos (2022) included CR providers, patients, and informal caregivers [16]. Sérvio (2019) included healthcare administrators, rehabilitation providers, and patients [19]. However, the context of CR in China remains unknown, let alone an examination from the perspectives of multiple stakeholders. Only four studies explored the situation in China: two included the perspective of patients [20,21], while the other two included the perspective of health professionals [12,22]. None of the studies considered simultaneously analyzing the context from multiple stakeholder perspectives.

Therefore, this study aimed to explore the barriers and facilitators of CR in China from the perspectives of multiple stakeholders and to provide potential suggestions for overcoming barriers and utilizing facilitators. We included hospital directors, health professionals, patients, and family members based on the previously mentioned literature [16,19]. In addition, we aimed to be comprehensive in our stakeholder selection; therefore, we further included rehabilitation lecturers and company employers. Hospital directors were included because they were responsible for implementing government policies and making decisions in their hospitals. Health professionals were included as they were direct healthcare providers. Patients were included due to their firsthand experience with CR, as they are the primary beneficiaries of the program. Family members were included because they were the informal caregivers. Additionally, we included rehabilitation lecturers who provided CR staff education and training and company employers who served as job providers. Company employers’ views on CR were also important, as some patients return to work after sick leave.

## 2. Materials and Methods

### 2.1. Interviewee Recruitment

This exploratory qualitative study was conducted using semi-structured interviews between March and June 2022. The interviewees were purposefully sampled from Tianjin and Xi’an to represent the perspectives of multiple CR stakeholders, including hospital directors, health professionals, cardiac patients, family members, rehabilitation lecturers, and company employers. Interviewee recruitment continued until saturation was reached, the point at which no new concepts emerged from the subsequent interviews [23,24].

### 2.2. Interview Guide

The interview guide was specifically designed (by K.Z., K.A., S.J., and Y.Z.) for each stakeholder group. The interview questions were developed based on existing literature, the experience of the research team, and the “Theory of Planned Behavior” [25].

Hospital directors were directly queried about their experiences and thoughts on barriers and facilitators of CR (“What barriers and facilitators can you think of for the implementation of rehabilitation centers?”), and suggestions for CR delivery (“What should be the proper way to include more patients?”; “How could the rehabilitation be reimbursed to maximize the number of attendants?”). In addition, questions regarding the sufficiency of CR staff were derived from previous literature [12,26], which pointed out the importance of this topic as a prerequisite for delivering CR services. Three interviews were conducted to test the interview guide for hospital directors. Following these test interviews, the guide was refined and finalized (see Appendix A).

For the others, we developed the interview guide (see Appendix A) using individual-level characteristics to describe the adoption of novel practices predicted by a person’s attitudes, social norms, and perceived control, which aligns with the “Theory of Planned Behavior (TPB)” [25]. This theory is used in the fields of medicine and nursing implementation research to identify the barriers and facilitators of innovation adoption [27]. Attitude refers to a person’s positive or negative evaluation of CR, such as whether they believe it helps with recovery. An example question is as follows: “Do you think CR will contribute to restore the functioning and health? Why or why not?” Social norms represent the influence of social pressures, including family, health professionals, or government, in shaping one’s decision to participate in CR. An example question is as follows: “Who is likely to convince patients to join CR (hint: government, doctors, families, friends, advertisement, live example of success CR)?” Lastly, perceived control involves an individual’s confidence in overcoming barriers, such as financial constraints, transportation difficulties, or doubts about effectiveness. An example question is as follows: “How accessible is the CR center for patients?” Besides TPB, some questions were developed based on the previous literature [16], e.g., the question for the patient and family member about the need for an accompanying person. Additional questions were formulated by the research team based on their expertise, such as “How do you motivate your colleagues (other health professionals) in the advancement of the rehabilitation program?” Then, the questions were reorganized and integrated into four general categories: (1) understanding of CR; (2) attitude towards health outcomes; (3) accessibility to healthcare; and (4) identifying potential strategies.

### 2.3. Data Collection

One-on-one interviews were conducted online (by Y.Z.) and lasted between 30 and 100 min. Field notes were taken during the interviews. Interviews were transcribed verbatim. The transcripts were returned to the interviewees for comments. To ensure that the non-Chinese-speaking research team could read and analyze the data, the transcripts were translated from Chinese to English (by Y.Z.). The translation was reviewed by a second researcher (J.D.). Differences in opinions were discussed, and the final decision was made with a third researcher (S.J.). All data were pseudonymized and treated as confidential. Written informed consent was obtained from each interviewee.

### 2.4. Data Analysis

Data were analyzed using thematic analysis in a six-phase analytic process: (1) familiarizing with the data, (2) coding, (3) developing themes, (4) reviewing potential themes, (5) defining and naming themes, and (6) producing the report [23]. The interview guide identified major thematic topics; therefore, the overall thematic topics were developed deductively. However, the inductive approach was applied for coding and subtheme development, under the assumption that coding develops depth and progresses through immersion and repeated engagement with the data [28].

The interviews were analyzed using ATLAS.ti [29]. For each stakeholder group, at least one interview was independently coded by two researchers (Y.Z. and J.D.). In case of disagreement, a discussion was held until a consensus was reached. The consensus criteria for coding were jointly established by the two researchers through iterative discussions and codebook refinement, ensuring alignment in the coding process. We aimed to code as many as possible to capture the depth and detail of the responses. Following the agreed criteria, the remaining interviews were coded by one researcher and checked by another. In addition, the codes were randomly checked and reviewed by a third researcher (S.J.). The codes were further modified based on the discussions of the three researchers. In the codebooks (see Appendix A), we left out the codes that were mentioned by only one interviewee or that did not belong to the subthemes.

To provide an overview, we categorized the barriers, facilitators, and potential suggestions into macro-, meso-, and micro levels. The macro-level concerns the health system, the meso-level concerns the delivery of CR, and the micro-level concerns individual perspectives.

The study was performed following the Consolidated Criteria for Reporting Qualitative Research (COREQ), a 32-item checklist [30] (see Appendix A). Because we conducted thematic analysis, a 15-Point Checklist of Criteria for Good Thematic Analysis Process [31] was also applied to check the quality of the process (see Appendix A).

## 3. Results

Fifteen CR stakeholders were interviewed: four hospital directors, four health professionals (two cardiothoracic surgeons, a physiotherapist, and a head nurse from the cardiology department), one cardiac patient, three family members, a rehabilitation lecturer, and two company employers. Other characteristics of the interviewees are presented in Table 1.

Three main thematic topics were developed based on the interview guide. Subthemes were developed through interviews. Table 2 presents the thematic topics and subthemes.

### 3.1. Barriers

#### 3.1.1. Lack of Resources

The lack of resources was identified as a major barrier to CR. According to Hospital director 1, there is a lack of financial resources and equipment.

The lack of financial reimbursement is the main aspect of the lack of resources. Three hospital directors and one employer indicated that CR costs were not covered by health insurance. Therefore, patients need to use their out-of-pocket money, as mentioned by Hospital director 3 and Health professional 4. According to several interviewees (two hospital directors, two health professionals, and two family members), this creates a cost burden for patients, which is a major barrier to attending CR. This is illustrated in the following quote:

“When I recommend CR to patients, they might not accept the suggestion due to various factors, including concerns about the costs associated with CR.”(Health professional 4)

Furthermore, CR costs are perceived as additional expenses by two family members. Family member 1 argued that “the surgery already costs a lot of money”; thus, the additional costs of CR would put an extra burden on patients. Family member 2 further explained: “For many families, it is already difficult for them to pay for the treatment of the disease. When you tell them to join CR, they are willing to participate, but they face financial constraints”.

#### 3.1.2. Lack of CR Professionals

The lack of CR professionals was mentioned as a barrier by all hospital directors and one rehabilitation lecturer. They believe that the number of CR professionals is insufficient. For example, Hospital director 1 indicated that “CR nurses are not enough”. Hospital director 3 said, “We lack rehabilitation therapists, psychologists, and nutritionists”.

The lack of CR professionals was explained by the fact that rehabilitation medicine was a young discipline in China, according to Hospital director 1. In addition, one hospital director and one rehabilitation lecturer pointed out that “the limited educational programs where staff can be trained” was another reason.

#### 3.1.3. Lack of Awareness and Acceptance

Lack of awareness was an essential barrier, according to the interviewees (one hospital director, two health professionals, and one employer). “Our people have just solved the problem regarding shortage of food and clothing, they may not even know about rehabilitation”, explained Employer 2.

In addition, hospital directors and health professionals believed that a lack of acceptance was another barrier. People do not accept CR since “they do not think rehabilitation is necessary/beneficial”, according to Hospital director 4.

Furthermore, interviewees pointed out that “CR is not considered as part of the treatment routine in China”. “Patients believe that they will recover to normal (health) levels after surgery, this is not true, they must learn to change their view on it”, said Health professional 1.

#### 3.1.4. Logistical Barriers

Several interviewees (one hospital director, one health professional, one patient, and one family member) also mentioned logistical barriers, including travel distance from home to the CR center, transportation to the CR center, and the need for an informal caregiver’s accompaniment.

“There are many factors that prevent them from participating CR, for example, the cost, the travel distance from the rehabilitation center, transportation to the rehabilitation center, and whether there is someone who can accompany them to the rehabilitation center. The person accompanying them is very important. For post-surgery patients, it may be a bit inconvenient to come by car or public transportation by themselves during the early postoperative period.”(Health professional 1)

#### 3.1.5. Lack of Coordination

Two hospital directors pointed out coordination barriers. “There is no integrated system between general hospital and the specialized rehabilitation institutions”, said Hospital director 3. As a result of a lack of coordination, physicians do not know where to refer patients to CR.

#### 3.1.6. Low Motivation of Certain Patients

According to one hospital director and three health professionals, certain patients have a relatively lower interest and are less motivated to attend CR, for example, patients with less obvious functional impairment, lower education level, poorer socioeconomic status, or older age.

In addition, patients who need to return to work take extra sick leave and incur wage loss if they spend extra time participating in rehabilitation programs; thus, this is a barrier for them to attend CR, according to Hospital director 4.

### 3.2. Facilitators

#### 3.2.1. Positive Attitude of CR Stakeholders

Support from health professionals is vital for promoting CR. All four health professionals expressed their willingness to promote CR in the eligible patients. Health professional 1 said, “We are concerned about the condition of patients after surgery […]. If rehabilitation is beneficial to the short- and long-term prognosis, we as physicians […] will certainly try to persuade patients to attend rehabilitation programs”.

The positive attitude of the patients was another facilitator. Patient 1 thought CR was beneficial in terms of improving physical strength and health condition and was willing to attend CR when advised by family members. In addition, Patient 1 thought that the patients themselves had an important role in regaining functioning and health situations through CR. This is essential since “for long-term, patients will need to manage their physical functioning and health by themselves”, said Health professional 1. In addition, Family member 2 held the opinion that a positive attitude would help patients to persist in CR training.

Family members were also positive towards CR, and they supported patients attending CR in several ways. For instance, one family member indicated that she tried to persuade the patient to join CR, and one patient mentioned that a family member took the initiative to consult health professionals about CR. Most importantly, the family members spent time accompanying patients attending CR.

#### 3.2.2. High Motivation of Certain Patients

According to the interviewees (two hospital directors, two health professionals, and one family member), certain patients have a relatively higher interest and are more motivated to participate in CR. For example, patients with relatively good socioeconomic status, higher expectations for quality of life, higher awareness, higher education level, young patients, or patients who suffered from functional impairment would be more likely to participate in CR.

#### 3.2.3. Increased Awareness

Increased awareness of CR was also identified as a facilitator. According to the interviewees, this is represented in two aspects: People’s awareness of CR is increasing, and the benefits of CR are gradually being seen by people.

As a result of increased awareness of CR, the demand for and attention to CR have been increasing as well. Hospital director 2 said that: “A lot of patients need rehabilitation. At least, in our hospital, we pay more attention to this issue”.

#### 3.2.4. Government and Policy Support

The government and national policy support of CR was a facilitator reported by two hospital directors and one lecturer. Hospital director 1 pointed out that a national policy was to promote three kinds of health centers, including the rehabilitation center (the other two were imaging centers and clinical testing centers), with the aim of making the services more accessible. Furthermore, the hospital director added that: “many provincial and municipal governments are trying to optimize the business environment to attract investment”, for example, “by simplifying the application process for obtaining a business license”. Therefore, this is a facilitator for setting up private CR centers.

#### 3.2.5. Financial Impact

The affordability of the CR price was identified as a facilitator based on the interviewees (three hospital directors, one patient, and two family members). Hospital director 3 said, “Even if it is at their own expenses, they (the patients) can afford it. […] The current prices are relatively low”.

#### 3.2.6. CR Services Becoming Available

Two hospital directors noticed that more CRs have become available in recent years. When the services become available and accessible, people gradually become aware of CR, as mentioned by Hospital director 1.

### 3.3. Suggestions Towards Implementing CR

#### 3.3.1. Increase the Awareness and Change Perspective

To overcome the barrier regarding the low awareness of CR, the interviewees (three hospital directors, two health professionals, one lecturer, and one employer) provided some suggestions. For example, through health education and the popularization of CR, patients can get to know about CR.

In addition, two hospital directors and one health professional thought that people needed to change their perspectives on CR. “They must recognize that rehabilitation is an integral part of overall disease treatment”, said Hospital director 2. The following quote further explains as follows:

“For example […], patients believe that they will recover to normal levels after surgery, this is not true. They must learn to change their perspectives towards it (CR).”(Health professional 1)

#### 3.3.2. Make CR Services Accessible

All hospital directors and one health professional suggested that establishing CR centers in secondary and primary health institutions (e.g., secondary rehabilitation hospitals and community health centers) to overcome the challenges of (1) the limited capacity for rehabilitation in tertiary hospitals and (2) the inconvenience of transportation to distant tertiary hospitals, thus making CR services more available and accessible.

#### 3.3.3. Establish Cooperation and Coordination

According to Hospital director 1, within a hospital, “nurses need to cooperate and coordinate with surgeons, specialist doctors, and rehabilitation therapists to ensure that the entire treatment process, including CR, is integrated”.

In addition, three hospital directors suggested encouraging cooperation and coordination between tertiary hospitals and community health centers to ensure smooth transition and continuous patient follow-up.

“We should establish the rehabilitation programs in the community or in the lower-level health institutions through Medical Alliance (cooperation). Patients can then participate in rehabilitation programs in their community instead of in tertiary hospitals. Rehabilitation experts from tertiary hospitals can provide guidance to staff working in community health centers.”(Hospital director 2)

#### 3.3.4. Develop Reimbursement Methods

The role of insurance in reducing the financial burden on patients and making CR affordable was emphasized by the interviewees (four hospital directors, one health professional, and one family member). Some suggested that National Health Insurance should reimburse CR costs, while others also mentioned the role of commercial insurance.

However, two hospital directors pointed out the situation that “most people in China rely on the National Health Insurance rather than commercial insurance”. To maximize the percentage of payment reimbursed by utilizing both National Health Insurance and commercial insurance, Hospital director 1 suggested that the rehabilitation items should be refined and categorized to “basic care” and “additional care”, then the National Health Insurance could cover the basic part while the commercial insurance could cover the additional part.

In addition, sometimes there is cooperation between enterprises and insurance companies. For instance, enterprises collectively buy commercial insurance for their employees, according to one hospital director and two employers. This could be a feasible way to cover the CR costs for patients with jobs.

#### 3.3.5. Motivate Stakeholders

The interviewees shared their opinions on motivating patients to participate in CR. Some (three hospital directors, one health professional, one patient, and two family members) believed that if the costs could be reimbursed by health insurance, more patients would attend CR. Furthermore, some (two hospital directors, two health professionals, one employer, one patient, and one family member) thought that it would be essential to inform patients about the benefits of CR. This can be accomplished through health education. Additionally, peer support was considered effective by two health professionals because it motivated patients to join and adhere to the CR program. Furthermore, Family member 1 mentioned that a successful rehabilitation example (of other patients) could motivate the patient to join CR.

Moreover, Hospital director 4 emphasized the importance of motivating health professionals. All four health professionals indicated that financial incentives would be helpful in motivating their colleagues (other health professionals) to refer or advise patients to CR.

### 3.4. Macro-, Meso-, and Micro-Level Barriers, Facilitators, and Potential Suggestions

Figure 1 illustrates the barriers, facilitators, and potential suggestions for CR implementation at the macro-, meso-, and micro-levels.

The macro-level concerns the health system. There is a lack of financial resources and integrated systems. The suggestion is to develop reimbursement methods and utilize the Medical Alliance (cooperation) system to integrate health services between tertiary hospitals and community health centers. Furthermore, CR should be included as part of the treatment routine, and eligible patients should be referred for CR after acute care.

The meso-level concerns the delivery of CR. Although CR services have recently become available, the overall number of CR programs is still insufficient. This is due to a lack of resources, including financial resources, education resources, staff, facilities, and equipment. In addition, resources are clustered mostly in tertiary hospitals. The suggestion is to establish CR in secondary hospitals or community health centers, while experts from tertiary hospitals could provide guidance to staff in lower-level health institutions.

The micro-level concerns individual perspectives. CR is affordable for some patients; however, it is paid by patients’ out-of-pocket money, which induces an extra financial burden on patients alongside the cost of surgery, which is commonly paid by patients as well. In addition, the vital barrier to CR is a lack of awareness, that is, people do not know about CR. Furthermore, people have not yet seen the benefits of CR, and they think that surgery is the end of treatment. Therefore, the possible solution is in two steps: The first step is to increase the awareness of CR among patients and health professionals through health education and popularization; the second step is to change their perspective, to inform them that surgery is not the end and CR is an essential part of the complete treatment process.

## 4. Discussion

This qualitative study revealed barriers to and facilitators of CR implementation in China from the perspectives of multiple stakeholders. According to the interviewees, the delivery of CR is impeded by a lack of resources, a shortage of CR professionals, and limited coordination between health institutions. Furthermore, CR participation is hindered by a lack of awareness, a lack of reimbursement, and insufficient access to CR. The facilitating factors of CR mentioned were a positive attitude of stakeholders, high motivation of some patients, policy support, and affordability of CR.

Our results are consistent with previous studies [11,16,19] on the underutilization of CR programs in other middle-income countries such as Iran, Colombia, and Brazil. The study in Iran surveyed patients, identifying transportation, cost, and insurance issues as major barriers [11]. The study in Colombia also surveyed patients, adding insights from informal caregivers and physiotherapists through focus groups, also highlighting financial and logistical barriers [16]. The study in Brazil included patients, healthcare administrators, and rehabilitation providers, revealing macro- and meso-level barriers like resource shortages and limited referrals [19]. Comparable to these studies, the interviewees reported that a lack of resources, insurance coverage of CR costs, transportation, and travel distance to the CR center were barriers. A study conducted in Brazil [19], for example, identified a lack of awareness as a barrier for patients to attend CR and for physicians to promote CR. This finding was also observed in the present study. The interviewees in this study offered several explanations: Surgery alone is often expected to be sufficient by patients, CR is not integrated into the treatment routine, and patients are reluctant to exercise after surgery. Hospital directors and health professionals have stressed the need for more health education to let patients and health professionals change their perspectives. Furthermore, people are likely to realize the importance of CR if the government decides that the cost is covered (in whole or in part) by insurance.

The lack of financial reimbursement was identified as a significant barrier to attending CR in both our study and previous studies [11,16,19,26]. Our study further revealed that CR cost was perceived as an additional expense on top of costly surgical or medical treatments, a concern not explicitly mentioned in previous literature. The reason behind this may be that in China, patients must pay 50% of the surgery fee out of pocket, whereas the majority of patients in Iran and Brazil receive surgery at little to no cost [32]. These financial barriers highlight the need for policy reforms, including improved insurance coverage and financial assistance programs.

This study identified a shortage of CR professionals as an essential barrier to CR delivery. This finding is in line with a review [33] regarding the barriers to CR in LMICs. The availability of CR professionals varies across countries [33]. According to our study, currently in China, there is a shortage of rehabilitation therapists, rehabilitation nurses, psychologists, and nutritionists. Furthermore, our results show that training programs for CR professionals are limited. Therefore, more training programs for CR professionals are required.

A positive perception of CR by healthcare directors and encouragement of patients from health professionals, family members, and other patients were seen in our study as facilitators, which is in line with the aforementioned study [16]. A unique finding from the current study was that patients with relatively good socioeconomic status, higher expectations for quality of life, higher awareness, higher education level, and younger age would be more likely to participate in CR. Additionally, government and policy support, as well as increased CR services, were identified as facilitators in our study. However, these subthemes were developed based on responses from the same population, which is a limitation of our study.

In addition, a WHO report addressed factors impacting global rehabilitation in general, including for individuals recovering from injury, surgery, disease, or age-related decline in functioning. Our findings align with this report, which identified out-of-pocket expenses, a shortage of trained rehabilitation professionals, limited resources, and ineffective referral pathways as barriers [34].

The setting of this study is in Tianjin and Xi’an, both are representatives of “new first-tier cities” and rank in the top 20 out of 337 cities in China in terms of commercial resources, urban hubs, activities of urban people, diversity of lifestyles, and future plasticity [35]. CR has been established in traditional “first-tier cities” such as Beijing and Shanghai. However, in some lower-level cities and rural regions, even cardiac surgery is not available, let alone CR. The uneven distribution of health resources, healthcare infrastructure, and human resources across the country causes health inequality, which has become a key issue that needs to be addressed [36,37].

This study has some limitations. Since we only included interviewees from two regions, the sample was not representative of the entire Chinese population. Also, hospital directors and health professionals with knowledge of rehabilitation were invited for the interview. It might also be possible that only patients and family members with a positive attitude towards rehabilitation accepted the interview. This may have biased the results towards an overly positive view of CR. In addition, the COVID-19 pandemic halted international travel; therefore, the interviews were conducted online, which might have led to shorter responses and less contextual information compared to face-to-face interviews [38]. Another impact of COVID-19 was that patient recruitment was challenging, as the interviewer was unable to conduct on-site recruitment to reach a larger number of patients. We acknowledge that the inclusion of only one patient is a limitation. Furthermore, we lack the perspective of the government, such as the national/regional Health Commission. This might have led to bias in the results regarding the macro-level barriers and facilitators. Finally, a potential limitation of this study is expectation bias, as neither the researchers nor the participants could be blinded. This may have influenced data collection and analysis. Efforts were made to minimize bias through peer debriefing, member checking, maintaining detailed documentation, and encouraging colleagues to review and challenge the research process and interpretations. Furthermore, we tried to ensure anonymity and confidentiality to create a safe environment. We believe that this reassured the interviewees that their responses would remain confidential and reduced social desirability bias.

To our knowledge, this is the first study to assess barriers and facilitators of CR in the Chinese context from the perspectives of multiple stakeholders and illustrate the barriers and facilitators at the macro-, meso-, and micro-levels. Furthermore, this study adds a unique patient perspective to current rehabilitation knowledge and encourages rehabilitation professionals to reflect on the ways and means of incorporating their perspectives into daily practice. Furthermore, the “semi-structured” nature of the interviews enabled us to get the insights of what the interviewees thought and experienced, thus allowing for the expounding information [39]. The findings of this study highlight key areas for improving CR accessibility and participation at multiple levels.

Compared to studies conducted in other countries, this study contributes several unique implications. First, our study specifically shows that the Medical Alliance (cooperation) system can help integrate CR services between the tertiary hospital and lower-level health institutions, as was mentioned by the hospital directors. Phase I inpatient CR can take place in tertiary hospitals, while patients can be referred for phase II CR to either a secondary rehabilitation hospital or a community health center closer to them. Experts from tertiary hospitals can develop an integrated CR plan and provide guidance to health professionals in lower-level health institutions. An important contribution of our study is that it makes clear that the Medical Alliance (cooperation) system can be utilized in other settings where resources are limited. Second, the high out-of-pocket cost of cardiac surgery makes CR an additional financial burden for patients in China, as noted by family members. More public funding is needed to cover the surgery and rehabilitation costs. Some enterprises collectively purchase commercial insurance for their employees, as mentioned by hospital directors and employers, offering a feasible way to finance CR. Third, CR awareness is very low in China, and surgery is often viewed as the final treatment step, as reported by hospital directors, health professionals, and employers. In high-income countries, CR is routinely included in treatment guidelines and widely implemented. However, in low- and middle-income countries (LMICs), the integration of CR into national treatment guidelines and its availability remain inconsistent. Our study reveals an urgent need in LMICs to raise awareness and ensure that policymakers integrate CR as an essential component of treatment guidelines, as seen in most high-income countries. Motivation for CR varies by socioeconomic status, education level, and age. Patients with better socioeconomic conditions, higher education, or greater awareness are more likely to participate in CR than those with less obvious functional impairment, lower education level, poorer socioeconomic status, or older age, according to hospital directors, health professionals, and family members. Targeted strategies are needed to include diverse patient groups. Lastly, rehabilitation medicine is still developing in China, with limited educational programs for CR staff, as noted by hospital directors and lecturers. More certified programs for CR staff are needed, similar to other LMICs. The International Council of Cardiovascular Prevention and Rehabilitation provides a certification program that trains students, community healthcare workers, and regulated health professionals to deliver care in low-resource settings [10]. Overall, significant efforts are required to advance the CR system in China. In the future, these insights can inform targeted interventions, such as policy reforms for CR reimbursement, expansion of CR services to secondary and primary health institutions, and enhanced patient education strategies. By addressing the barriers, facilitators, and potential strategies, the study contributes to the long-term goal of increasing CR utilization, improving patient outcomes, and reducing the burden of cardiovascular disease in China and potentially other middle-income countries.

Future research could further explore these findings using the Chinese/Mandarin Cardiac Rehabilitation Barriers Scale (CRBS-C/M) [40] to assess barriers on a larger scale. To enable a broader delivery of CR, future research should also explore the CR context in lower-level cities and rural areas. Additionally, the perspectives of patients and health professionals with non-positive attitudes toward CR should be taken into account for further research.

## 5. Conclusions

At the micro-level, more awareness regarding the effectiveness and benefits of CR is needed. Currently, patients believe that surgery alone is sufficient. Health education is required to change their views on CR. At the meso-level, implementing CR in secondary and primary health institutions could overcome barriers regarding travel distance and transportation to faraway hospitals. At the macro-level, CR reimbursement methods are needed to ease the financial burden on patients.

This study provides guidance for policymakers to enhance the delivery of CR services in China and potentially other middle-income countries.

## Figures and Tables

**Figure 1 ijerph-22-00574-f001:**
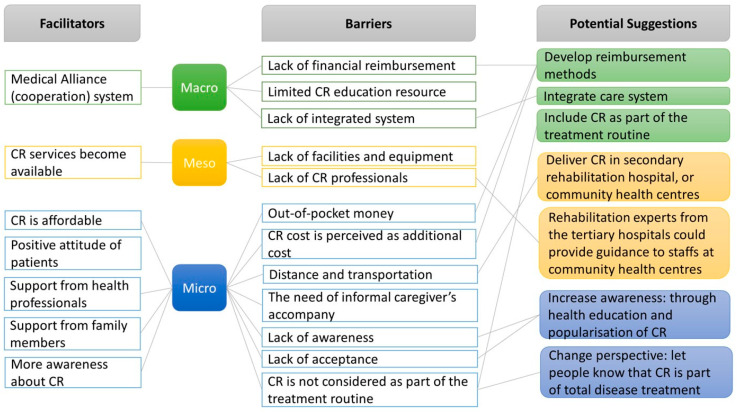
Macro-, meso-, and micro-level barriers, facilitators, and potential suggestions.

**Table 1 ijerph-22-00574-t001:** Characteristics of the interviewees.

	Hospital Director	Health Professional	Patient and Family Member	Lecturer and Company Employer
Number of interviewees [N]	4	4	4	3
Mean age [years, (SD)]	39.8 (6.7)	35.8 (0.5)	38.8 (3.3)	40.7 (6.0)
Gender [N]				
- Men	3	2	1	2
- Women	1	2	3	1
Education [N]				
- Master/PhD	4	2		
- University/College		2	1	3
- Middle/Secondary School			3	
Region [N]				
- Tianjin	2	2	4	3
- Xi’an	2	2		

SD: standard deviation.

**Table 2 ijerph-22-00574-t002:** Thematic topics and subthemes.

Thematic Topics	Subthemes
1. Barriers	Lack of resources
	Lack of CR professionals
	Lack of awareness and acceptance
	Logistical barriers
	Lack of coordination
	Low motivation of certain patients
2. Facilitators	Positive attitude of CR stakeholders
	High motivation of certain patients
	Increased awareness
	Government and policy support
	Financial impact
	CR services becoming available
3. Suggestions towards implementing CR	Increase awareness and change perspective
	Make CR services accessible
	Establish cooperation and coordination
	Develop reimbursement methods
	Motivate stakeholders

## Data Availability

The raw data supporting the conclusions of this article will be made available by the authors on request.

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
