# Peer review of "Barriers and Facilitators of Cardiac Rehabilitation in a Middle-Income Country: A Qualitative Study from China"

_ijerph, 2025, doi:10.3390/ijerph22040574_

Round 1
Reviewer 1 Report
Comments and Suggestions for Authors
This paper has comentarios based on the revision that was done by the evaluator

Reviewer 2 Report
Comments and Suggestions for Authors
All the comments are in the file attached.

Reviewer 3 Report
Comments and Suggestions for Authors
This study identified the barriers and facilitators of cardiac rehabilitation in China through an exploratory qualitative study called semi-structured interviews. Semi-structured interviews are a commonly used qualitative research method in the social sciences. It's an interesting topic.
However, there are some problems with the study. Why did the author interview 15 people instead of more or less? How do you know if the sample size is sufficient? Did you do the theoretical saturation test? Did you make a saturation table? Anyway, by other methods. In this study, four hospital directors,four health professionals,one cardiac patient, three family members, one rehabilitation lecturer and two company employers were interviewed. Why did the numbers vary? Because of the nature of qualitative research, subjects and researchers were not blinded. Therefore, they may be subject to expectation bias. This should be included in the limitations. The sample size should also be included in the limitation. What is the value and significance of the results of the article for the future?
Round 2
Reviewer 2 Report
Comments and Suggestions for Authors
Dear Authors,
Thank you for the opportunity to read again and review your manuscript.
All the comments that I suggested in the previous review phase were addressed and now the paper sounds more suitable for a scientific publication.
One aspect that could be better clarified is related to the questionnaire development. The authors added that the provided questionnaire derived from the existing literature, the experience of the research team and the “Theory of Planned Behavior”. However, it is not clear how the different aspects were integrated: only the questions derived from the TPB are specified but details regarding how the questions are derived from the literature or from the researchers’ experience are not provided.
Another comment, that I would provide to the Authors, is related to the discussion. I really appreciated the effort dedicated to improve this section. However, I suggest to the authors to emphasize the contribution of the research. If the study is the first study to assess barriers and facilitators of CR in the Chinese context from the perspectives of multiple stakeholders, which are the difference emerged in this specific geographical context in comparison with other geographical or territorial contexts?
I think that, after these integrations, the paper will be ready for the publication!
Good luck!
